# Synthesis and Hydration Characteristic of Geopolymer Based on Lead Smelting Slag

**DOI:** 10.3390/ijerph17082762

**Published:** 2020-04-16

**Authors:** Liwei Yao, Degang Liu, Yong Ke, Yuancheng Li, Zhongbing Wang, Jiangchi Fei, Hui Xu, Xiaobo Min

**Affiliations:** 1School of Metallurgy and Environment, Institute of Environmental Science and Engineering, Central South University, Changsha 410083, China; yaoliwei0125@126.com (L.Y.); keyong000ke@csu.edu.cn (Y.K.); LYC_DU@163.com (Y.L.); wzbing@126.com (Z.W.); jack-fei@csu.edu.cn (J.F.); Xuhui940303@163.com (H.X.); 2School of Metallurgical and Chemical Engineering, Jiangxi University of Science and Technology, Ganzhou 341000, China; liudegang6669@163.com; 3Chinese National Engineering Research Center for Control and Treatment of Heavy Metal Pollution, Changsha 410083, China

**Keywords:** lead smelting slag, geopolymer, iron behavior, immobilization, hydration characteristic

## Abstract

Lead smelting slag (LSS) has been identified as general industrial solid waste, which is produced from the pyrometallurgical treatment of the Shuikoushan process for primary lead production in China. The LSS-based geopolymer was synthesized after high-energy ball milling. The effect of unconfined compressive strength (UCS) on the synthesis parameters of the geopolymer was optimized. Under the best parameters of the geopolymer (modulus of water glass was 1–1.5, dosage of water glass (W(SiO_2_+Na_2_O)) was 5% and water-to-binder ratio was 0.2), the UCS reached 76.09 MPa after curing for 28 days. The toxicity characteristic leaching procedure (TCLP) leaching concentration of Zn from LSS fell from 167.16 to 93.99 mg/L after alkali-activation, which was below the limit allowed. Meanwhile, C-S-H and the geopolymer of the hydration products were identified from the geopolymer. In addition, the behavior of iron was also discussed. Then, the hydration process characteristics of the LSS-based geopolymer were proposed. The obtained results showed that Ca^2+^ and Fe^2+^ occupied the site of the network as modifiers in the glass phase and then dissociated from the glass network after the water glass activation. At the same time, C-S-H, the geopolymer and Fe(OH)_2_ gel were produced, and then the Fe(OH)_2_ was easily oxidized to Fe(OH)_3_ under the air curing conditions. Consequently, the conclusion was drawn that LSS was an implementable raw material for geopolymer production.

## 1. Introduction

China has been the largest producer and consumer of lead in the world for years [1,2]. At present, primary lead production in China is performed via traditional sintering blast furnace smelting, the Shuikoushan process, the Isa smelt system, the Kaldo converter lead smelting process and the Queneau-schuhmann-lurgi lead smelting process [3,4]. Among them, the Shuikoushan process is the main primary lead smelting technology in China, due to its advantages of being energy saving and having a higher metal recovery and longer furnace life [1,5]. In lead smelting, the molten slag is then commonly treated by water quenching to obtain lead smelting slag (LSS), which contains a high content of iron, silicon, calcium and aluminum oxides [6]. For each ton of metallic lead production, 100–350 kg of LSS is generated [7,8]. As a result, a huge amount of LSS is generated from primary lead production. The LSS contains quantities of minor and trace heavy metals [9,10]. It can contaminate the environment through leaching if it is not constrained [11,12,13,14]. In consequence, how to deal with LSS is a serious problem for lead smelters.

Traditionally, LSS was either recycled back into the smelting process or disposed of in piles on site [15]. In recent years, LSS has been used as an aggregate in concrete production [16] and also used in value-added streams, such as in cement clinker production and heavy clay production [17,18,19,20]. Additionally, the immobilization of LSS with a coal fly ash-based geopolymer has been investigated in some recent studies [21,22]. However, the consumption of LSS is limited in these treatments. Therefore, a method that can increase the value-added of IRLSS as well as deal with it in large capacity needs to be developed.

Unlike other applications, the production of a geopolymer primarily based on LSS is possible. The geopolymer has emerged as an alternative to ordinary Portland cement owing to its superior durability and environmental performance [23]. Because of these advantages, the geopolymer has found a variety of applications, such as transportation, industrial, agricultural, residential and mining [24,25,26]. Besides these, one of its major newer applications is in waste management, especially in the immobilization of toxic metals, such as Zn, Cu, Cd, Cr and Pb [27,28,29]. A geopolymer is defined as a synthetic alumino–silicate material and is generated from the reaction of solid alumino-silicate with a highly concentrated aqueous alkali hydroxide or silicate solution [17]. A number of materials have been investigated as candidates for geopolymer production, such as blast furnace slag [30], metakaolin [31], fly ash [32], kaolinitic clays [33], municipal solid waste incineration fly/bottom ashes [34,35] and red mud [36]. However, to our knowledge, there are few correlated studies concerning the geopolymer based on LSS.

Geopolymers are inorganic polymers and are constituted of alternating SiO_4_ and AlO_4_ tetrahedra chains connected by a shared oxygen atom and balanced by cations [37]. The presence of iron can substitute for Al and may play important roles in the structure and properties of geopolymers [38,39]. In our previous study, LSS was pretreated as a geopolymer precursor through the high-energy ball milling activation process. It could be used as a geopolymeric solidification/stabilization (S/S) reagent for municipal solid waste incineration fly ash (MSWI FA) [40]. However, to our knowledge, LSS contains a great amount of iron and the behavior of iron in the LSS based geopolymer has not yet been studied. It is necessary to explore the possibility of LSS as a high-performance alkali-activated slag-based cement.

In the present work, the LSS was mixed with water glass for a geopolymer production with a large consumption of LSS. The unconfined compressive strength (UCS) was optimized and the toxicity characteristic leaching procedure (TCLP) was performed. The hydration products were analyzed with x-ray diffractometry (XRD), Fourier transform infrared (FTIR) and Mössbauer. Finally, the hydration process characteristics of the LSS-based geopolymer were proposed and the behavior of iron was discussed.

## 2. Materials and Methods

### 2.1. Materials

The LSS used in the experiments was obtained from a lead smelting company in the south of China. The LSS was dried to a constant weight (±0.1 g) at 105 °C. The LSS was milled under the ball-to-material weight ratio of 5 in planetary ball milling at 400 rpm for 3 hours and passed through a 45 µm mesh sieve [41]. The particle size of the LSS powder was analyzed by laser granulometry. As shown in Figure 1, the LSS powder had a particle size of between 0.92 and 41.84 µm. The size of the particles was distributed in three concentrated areas of approximately 1.18, 4.32 and 13.50 μm. The median particle size (D50) of 5.28 μm indirectly reflected that some small particles might be agglomerated into large particles. The chemical compositions of the LSS (Table 1) were detected using x-ray fluorescence (XRF).

The water glass (SiO_2_ = 26.5%, Na_2_O = 8.3%, molar ratio of SiO_2_/Na_2_O is 3.3 and density is 1.371 g/cm^3^) was provided by Shandong Usolf Chemical Technology Co., Ltd. Other chemicals were analytical grade and were purchased from Sinopharm Chemical Reagent Co., Ltd.

### 2.2. Experimental Procedure

The milled LSS, water glass and deionized water were mixed and stirred evenly in proportion. Then, the slurry was poured into a steel mold (20 × 20 × 20 mm), uniformly shaken, covered with a plastic film and then cured for 24 hours in a cement concrete standard curing box at a 95% relative humidity at 20 ± 2 °C. The matrices were demolded and cured again for different time under the same conditions. Afterwards, the unconfined compressive strength (UCS) of the matrices was measured after curing for 3 and 28 days. 

In the experiment, the UCS was taken as an index to optimize the optimal parameters of the LSS-based geopolymer. The influences of the modulus (Ms) of water glass, dosage of water glass (W_Na2O + SiO2_) and water-to-binder ratio on the UCS were investigated. The formulation design of the LSS-based geopolymer experiments is presented in Table 2.

### 2.3. Tests 

#### 2.3.1. UCS Test 

The UCS tests were performed according to the GB/T17671-1999. The UCS tests of each sample were performed on three cubes as parallel experiments. All the samples were tested after curing for 3 and 28 days. The UCS of the matrices was tested using an unconfined compression machine (TYA-300B, Wuxi Xinluda Instrument Co., Ltd., Wuxi, China) with a loading rate of 2.4 kN/S. 

#### 2.3.2. Leaching Test

The TCLP was used to evaluate the leaching ability of heavy metals for the samples. An acetic acid solution with a pH of 2.88 ± 0.02 was selected as the leaching solution [42]. Amounts of 3.0 g of the crushed matrices (<9.5 mm) and 60 mL of the leaching solution were poured into the sealed polyethylene vessels and then shaken on a shaker with a speed of 30 rpm for 18 h. The leachates were filtered with a 0.45 µm membrane filter. Finally, the heavy metals concentrations in the filtrates were analyzed using an inductively coupled plasma-atomic emission spectroscopy (ICP-AES, IRIS Intrepid II XSP). All of the above experiments were carried out in triplicate and the results were calculated to obtain the average values.

#### 2.3.3. Other Tests

The XRF can provide qualitative and semi-quantitative analyses of elements in solid samples. A sample of the LSS was ground to ensure that the particle size was less than 45 µm and then analyzed by XRF (S4-Pioneer, Bruker Ltd., Karlsruhe, Germany). The crystallographic composition of the samples was characterized by x-ray diffraction (XRD, D/max2550 VB + 18 KW) at a speed of 10° min^−1^ in a 2θ range from 10° to 70°. ^57^Fe Mössbauer spectra were collected in a standard transmission geometry using a standard constant acceleration EG&G spectrometer with a ^57^Co(Rh) source. Measurements were performed with a constant acceleration at 20 °C and the calibration was referenced to metallic iron foil. Absorbers were prepared with 1.5 g of the sample powder in a lead sample holder.

## 3. Results and Discussion

### 3.1. Strength Optimization of LSS-Based Geopolymer

#### 3.1.1. Effect of Modulus of Water Glass

The effects of the modulus of water glass on the UCS of the LSS based geopolymer are shown in Figure 2. As observed in Figure 2, the UCS was a non-monotonous function of the modulus of water glass. The UCS of the geopolymer increased as the modulus of water glass increased from 0.5 to 1, and reached a maximum value of 73.6 MPa after hydrating for 28 days. Further increasing the modulus of water glass to 2 resulted in the UCS decreasing remarkably. This is because OH**^−^** is used for leaching soluble Si and Al from slag [43]. Moreover, the dissolution rate of Si and Al increases with the increase in OH**^−^** concentration. Soluble Si is essential for C-S-H and geopolymer gels production. Therefore, increasing the modulus of water glass is equivalent to increasing the content of soluble Si per volume. However, when the modulus of water glass was excessively low, the NaOH in the aqueous solution was superfluous and the frost phenomenon appeared easily. The frost phenomenon could result in many large voids in the geopolymer [44] and a lower UCS. In conclusion, the optimum modulus of water glass was 1–1.5.

#### 3.1.2. Effect of Dosage of Water Glass

Figure 3 presents the effect of the water glass (W(SiO_2_+Na_2_O)) dosage on the UCS of the LSS-based geopolymer. The results show that the UCS increased drastically from 2.32 to 74.45 MPa as the dosage of water glass increased from 2% to 5%. Then, the UCS decreased slowly as the dosage of water glass increased from 5% to 14%, because the concentration of OH^-^ increased as the dosage of water glass increased and accelerated the hydration degree of the LSS. Therefore, increasing the water glass dosage could increase the amount of gels per volume for geopolymer production. This resulted in the UCS of the geopolymer increasing. However, when the dosage of water glass was excessive, the frost phenomenon also occurred, the same as it did when the modulus of water glass was excessively low. It became more serious when the dosage of water glass and curing time increased. Therefore, the UCS of the LSS-based geopolymer cured for 28 days was lower than that of the geopolymer cured for 3 days, and an optimum modulus of water glass of 5% was selected.

#### 3.1.3. Effect of Water-To-Binder Ratio

As shown in Figure 4, the effect of the water-to-binder ratio on the UCS of the LSS-based geopolymer was studied in the range of 0.175 to 0.275. The UCS decreased almost linearly as the water-to-binder ratio increased. According to the results, the optimum water-to-binder ratio was 0.2. Using this value, the UCS of the LSS based geopolymer reached 76.09 MPa after curing for 28 days. Essentially, the geopolymer matrix consisted of two solid phases after alkali-activation (i.e., non-dissolved granules of the LSS and gels). When the water-to-binder ratio increased, the concentration of OH**^−^** was reduced, resulting in both the hydration degree of the LSS and the gels per volume of the produced geopolymer being reduced. In addition, the porosity of the matrix also increased. Due to these two aspects, the UCS of the geopolymer decreased when the water-to-binder ratio increased.

### 3.2. Hydration Characteristics Analysis

#### 3.2.1. Hydration Products Analysis

The XRD patterns of LSS and the LSS based geopolymer (R_2_) cured for 28 days are shown in Figure 5. The initial LSS pattern mainly consisted of an amorphous phase with some crystalline phases of predominantly magnetite (Fe_3_O_4_) and a little wuestite (Fe_0.872_O). The broad and diffuse peaks from the initial LSS around 28–35^o^ (2*θ*) reflected the short-range order of the CaO–Al_2_O_3_–MgO–SiO_2_ glass structure [45]. This is a common feature of an amorphous phase and is an indication of a rather reactive phase. 

The XRD pattern of the LSS based geopolymer was slightly different from that of the initial LSS. The broad and diffuse peaks for the geopolymer at around 28–35^o^ (2*θ*) became wider than those for the initial LSS, indicating that the geopolymer was generated. In addition, some weak peaks of calcium silicate hydroxide (C-S-H) were detected. The results were consistent with those of Li et al. [46,47,48]. Meanwhile, a new phase of iron hydroxide also appeared. However, the peak intensities of magnetite and wuestite were almost unchanged in the LSS based geopolymer compared with the initial LSS, indicating that magnetite and wuestite may not take part in the hydration reaction.

The conclusions of the XRD were strengthened by the analysis of the FTIR (Figure 6). The spectrum of the initial LSS was composed of a broad band at 946 cm^−1^, ascribed to ν_3_(Si–O) stretching modes, and another located at 507 cm^−1^, which was assigned to ν_4_(O–Si–O) bending modes of the SiO_4_ tetrahedral. 

For the LSS-based geopolymer, the spectrum had a sharp and intense absorption band at 1455 cm^−1^, attributed to the stretching vibrations of O–C–O bonds [43]. This absorption band indicated the presence of carbonates. Calcium carbonate could be the main carbonate compound, mainly because calcium ions in the pore fluid in the hardened body easily react with CO_2_ in the air to form CaCO_3_ during the curing process. The ones referred to as asymmetric stretching Si–O–Si vibrations (970–1090 cm^−1^) comprise the major sign of geopolymerization, according to Panias et al. [43]. The band at 978 cm^−^^1^ was attributed to the ν_3_(Si–O) stretching vibration in the geopolymer and was narrower and higher than anhydrous slag, indicating that the polymerization degree of the Si-O bond increased and that a short-range order was formed in the structure of the geopolymer. This was associated with the ν_3_(Si–O) stretching vibrations in geopolymer gel [49,50]. Finally, the absorption band was observed in the spectrum of the geopolymer at the wave number 466 cm^−1^. This was assigned to the ν_4_(O–Si–O) band and attributed to the formation of the C-S-H phase [51]. 

#### 3.2.2. Iron Behavior Analysis

Figure 7a shows the Mössbauer spectrum of the initial LSS at room temperature. It consisted of two distinct isomer shifts (IS) and quadrupole splitting (QS). One was equal to Fe^3+^(IS = 0.78 mm/s and QS = 1.21 mm/s). The value of the IS may be characterized by the Fe^3+^ with distorted tetrahedral symmetry [52,53]. The QS value of Fe^3+^ was a little larger than 1.2 mm/s, indicating the presence of a larger distortion of FeO_4_ tetrahedra. The linewidth value of iron oxide presented in a glass network is generally larger than 0.4 mm/s [52,53]. The linewidth values of Fe^3+^ and Fe^2+^ were 0.9 and 0.64 mm/s from the LSS, respectively, indicating that partial iron was present in the glass phase. The IS value of Fe^2+^ was 1.02 mm/s, indicating a distorted octahedral symmetry [52]. The QS value (1.99 mm/s) of Fe^2+^ was much smaller than those for typical distorted tetrahedral Fe(II). It may be that the Fe^2+^ occupied the site of the network as a modifier, as well as Ca^2+^, Na^+^, K^+,^ etc. The LSS contained many Ca and Fe but the content of Na and K was low, according to the XRF analysis (Table 1). Therefore, Fe and Ca became the main network modifier in the glass phase.

The Mössbauer spectrum of the LSS based geopolymer is shown in Figure 7b. The variation of the distortion of FeO_4_ tetrahedra was embodied in the QS. It can be seen that the QS values of Fe^2+^ decreased from 1.99 to 1.59 mm/s. These smaller QS values of Fe^2+^ indicated a lessened distortion of the Fe(II)O_4_ tetrahedra, which showed that Fe^2+^ dissociated from the glass phase and Fe(OH)_2_ appeared. The structural relaxation was embodied in the linewidth, as can be seen in Table 3. The linewidth values of Fe^3+^ in the LSS were reduced from 0.9 to 0.28 mm/s after the alkali-activation. The decrease in linewidth indicated that the homogeneity of the Fe–O bond length and O–Fe–O bond angle increased for the weak crystal phase of Fe(OH)_3_. This dissociated Fe^2+^ from the glass network and caused it to be oxidized to Fe^3+^ under the air conditions.

By comparing the phase changes between the LSS and LSS-based geopolymer and analyzing the iron behavior, we have deduced that the hydration process characteristics could be described as below. The Ca–O bond and Fe–O bond were broken in the glass network phase under the action of OH^-^ to form Ca(OH)_2_ and Fe(OH)_2_, respectively, because the bond energy between the network modifier and oxygen was lower than others were. Meanwhile, the [SiO_4_] was depolymerized from high degree polymerization to a lower one and became a monomer in the end. A small amount of high degree polymerization of [AlO_4_] was also depolymerized to a monomer in the same way as [SiO_4_]. Then, the C-S-H and geopolymer gels were produced. However, Fe(OH)_2_ was not detected in the LSS-based geopolymer and some weak peaks of Fe(OH)_3_ were detected. The reason for this was that the Fe(OH)_2_ was easily oxidized to Fe(OH)_3_ under the air curing conditions.

### 3.3. Heavy Metals Solidification 

Table 4 lists the concentrations of the main heavy metals of leachates from the initial LSS and geopolymer (R_2_) cured for 28 days via the TCLP test. Compared with the initial LSS, Hg, Ag and Se were also not detected in the leachate. The concentrations of Be, Cr and Pb did not change substantially and all were far below the limits allowed. The concentrations of Cd and Ba decreased significantly and were below the limits allowed. Moreover, the leachate concentration of Zn for R_2_ was 93.99 mg/L, which was slightly below the limit allowed. The arsenic concentration in the leachate for R_2_ was below the limits and increased slightly compared with the initial LSS. Therefore, we can conclude that the LSS-based geopolymer has a certain capacity for the immobilization of Zn.

C-S-H, the geopolymer and Fe(OH)_3_ have a good ability to immobilize Zn, Cd, Ni, As, etc. [54,55,56,57,58,59]. Liu et al. [60] reported that the immobilization capacity of C-S-H for Cu^2+^ and Cd^2+^ was better than for Zn^2+^, and that the low Ca/Si of C-S-H could fix Zn^2+^ better than the high Ca/Si of C-S-H. Therefore, the immobilization effect was better for Cu^2+^ and Cd^2+^ than for Zn^2+^. The Ca/Si of C-S-H in the LSS-based geopolymer was about 0.8, which suggested a good adsorption of Zn^2+^. Due to the combined actions of these hydration products, Zn and other heavy metals were immobilized effectively.

## 4. Conclusions

High-energy ball milling was used to activate the potential water-hardness properties of LSS. Afterwards, the LSS-based geopolymer was synthesized via water glass activation. The effect of UCS on the synthesis parameters of the LSS-based geopolymer was optimized. Under the best parameters of the binder (modulus of water glass was 1-1.5, dosage of water glass (W(SiO_2_ + Na_2_O)) was 5% and water-to-binder ratio was 0.2), the UCS reached 76.09 MPa after curing for 28 days. Meanwhile, the TCLP leaching concentration of Zn from the LSS fell from 167.16 to 93.99 mg/L after the alkali-activation, which was a significant reduction and slightly below the limit allowed.

Moreover, the C-S-H and geopolymer phases of the hydration products were identified in the LSS-based geopolymer. The behaviors of iron were also discussed. Then, the hydration process characteristics were proposed. The results indicated that calcium and partial iron performed as network modifiers in the glass network phase. The glass phase was dissociated with water glass and then Ca^2+^ and Fe^2+^ were dissolved to produce C-S-H, the geopolymer and Fe(OH)_2_. However, the Fe(OH)_2_ was easily oxidized to Fe(OH)_3_ under the air curing conditions. In consequence, we can draw the conclusion that LSS is an implementable raw material for geopolymer production or high-performance alkali-activated cement. 

## Figures and Tables

**Figure 1 ijerph-17-02762-f001:**
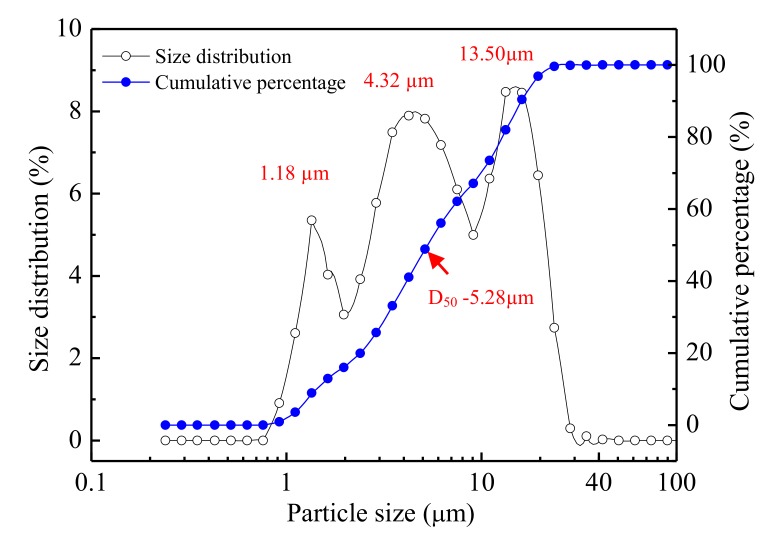
Particle size distribution of the lead smelting slag (LSS) after ball milling.

**Figure 2 ijerph-17-02762-f002:**
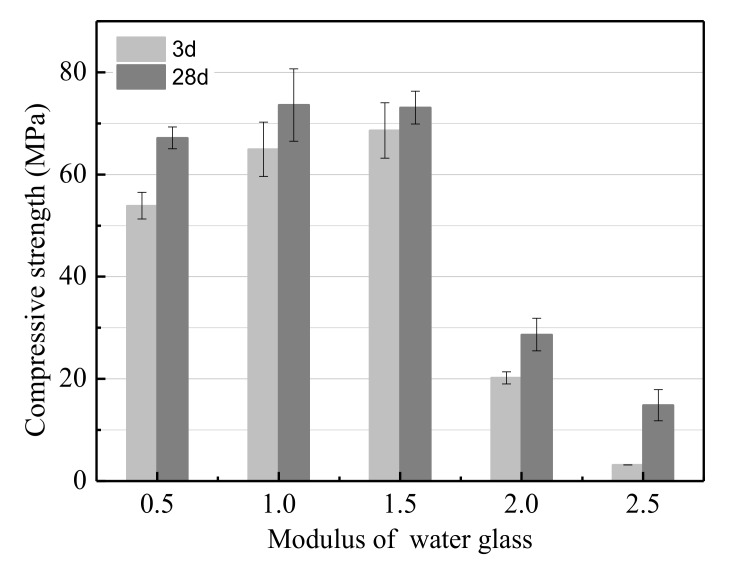
The effect of the modulus of water glass on the unconfined compressive strength (UCS).

**Figure 3 ijerph-17-02762-f003:**
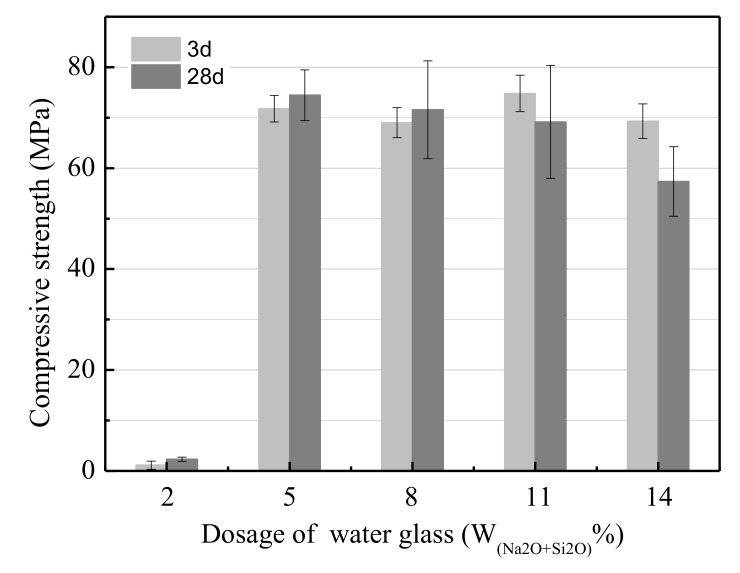
The effect of the water glass dosage on the UCS.

**Figure 4 ijerph-17-02762-f004:**
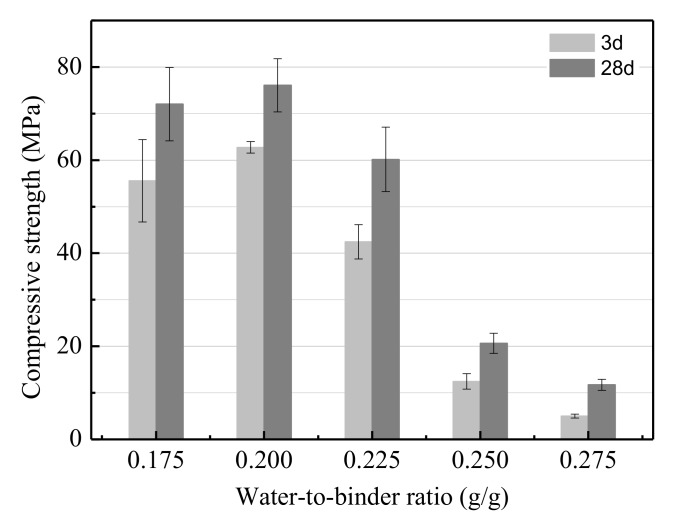
The effect of the water-to-binder ratio on the UCS.

**Figure 5 ijerph-17-02762-f005:**
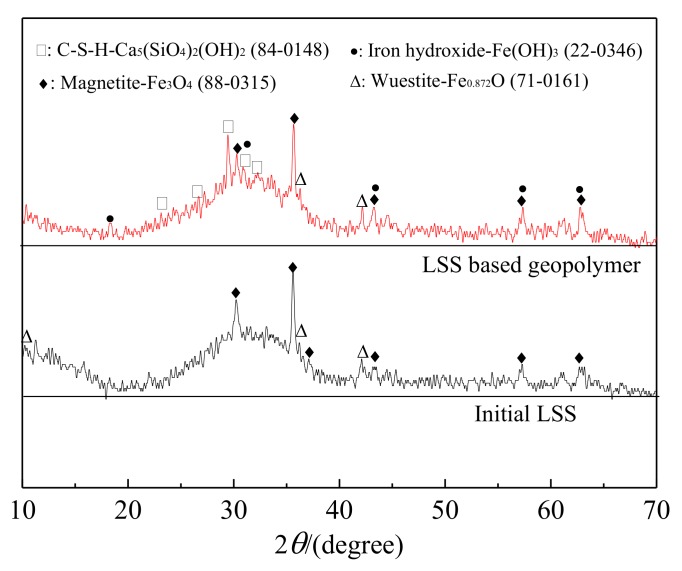
XRD patterns of the initial LSS and LSS based geopolymer (R_2_) after being hydrated for 28 days.

**Figure 6 ijerph-17-02762-f006:**
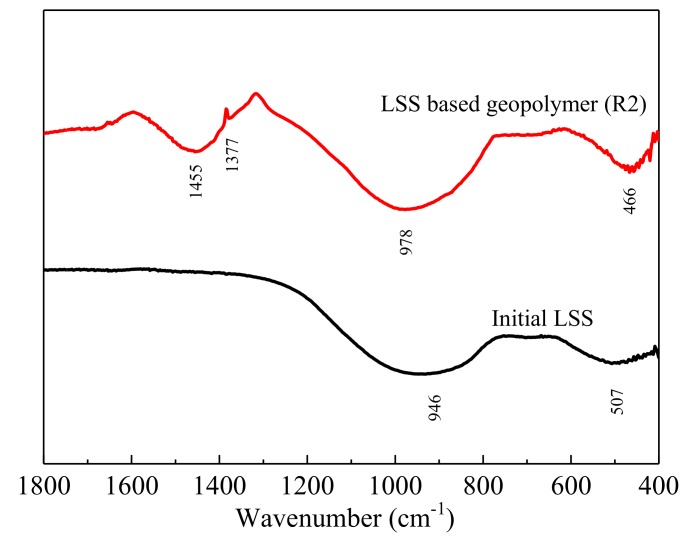
FTIR patterns of the initial LSS and LSS based geopolymer (R_2_) after being hydrated for 28 days.

**Figure 7 ijerph-17-02762-f007:**
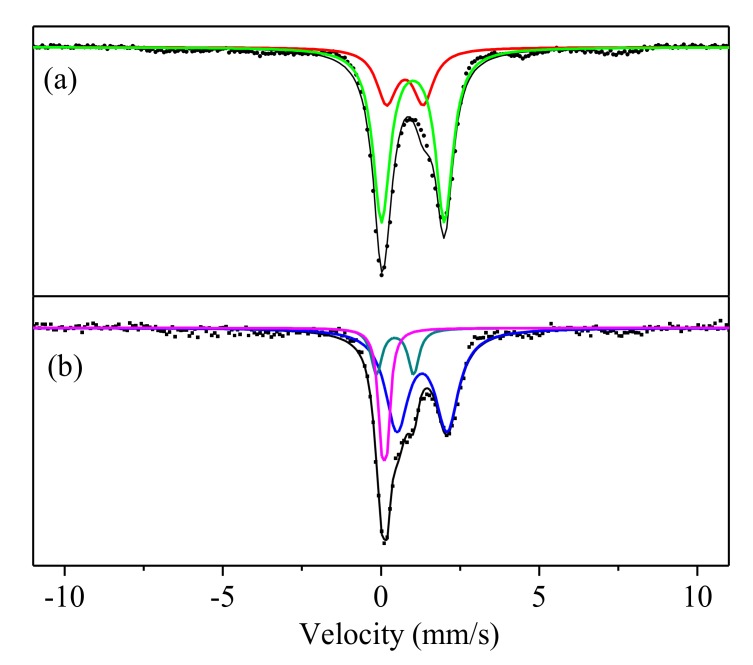
Mössbauer spectrums of the initial LSS (**a**) and LSS based geopolymer (R_2_) after being hydrated for 28 days (**b**).

**Table 1 ijerph-17-02762-t001:** Chemical compositions of the LSS (wt.%).

Element	Fe_2_O_3_	SiO_2_	Al_2_O_3_	CaO	MgO	ZnO	MnO_2_	Na_2_O	Cr_2_O_3_	CuO	PbO	As_2_O_3_	NiO_2_
LSS	23.01	32.23	5.51	24.58	4.54	5.19	2.06	1.62	0.29	0.32	0.08	0.02	0.17

**Table 2 ijerph-17-02762-t002:** The formulation design of the LSS-based geopolymer experiments.

Item	LSS /wt.%	W(SiO_2_+Na_2_O) /wt.%	Ms (SiO_2_/Na_2_O)	Water-To-Binder (g/g)
R_1_	95	5	0.5	0.2
R_2_	95	5	1	0.2
R_3_	95	5	1.5	0.2
R_4_	95	5	2	0.2
R_5_	95	5	2.5	0.2
W_1_	98	2	1	0.2
W_2_	92	8	1	0.2
W_3_	89	11	1	0.2
W_4_	86	14	1	0.2
M_1_	95	5	1	0.175
M_2_	95	5	1	0.225
M_3_	95	5	1	0.25
M_4_	95	5	1	0.275

Notice: binder = LSS + W_SiO2 + Na2O_ = 100%.

**Table 3 ijerph-17-02762-t003:** Mössbauer parameters of the initial LSS (a) and LSS based geopolymer (R_2_) after being hydrated for 28 days.

Sample	IS(mm/s)	QS(mm/s)	<H>(kOe)	RA(%)	Linewidth(mm/s)	Specie
Initial LSS (a)	0.78	1.21	-	32	0.90	Fe^3+^
	1.02	1.99	-	68	0.64	Fe^2+^
LSS based geopolymer (R_2_) *(b)*	0.11	0.17	-	19.7	0.28	Fe^3+^
	0.44	1.17	-	14.5	0.28	Fe^3+^
	1.30	1.59	-	65.8	0.28	Fe^2+^

**Table 4 ijerph-17-02762-t004:** Toxicity characteristic leaching procedure (TCLP) results of the LSS and geopolymer (R_2_) cured for 28 days.

Element	Zn	Cd	Ni	Cu	As	Ba	Be	Cr	Pb	Ag	Se	Hg
Limits	100	1	5	100	5	100	0.02	5	5	5	1	0.1
LSS	167.16	0.22	0.45	0.07	0.05	12.53	0.01	0.01	0.15	ND	ND	ND
geopolymer (R_2_)	93.99	0.07	0.11	0.02	0.09	4.72	0.01	0.01	0.15	ND	ND	ND

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
