# Peer review of "Synthesis and Hydration Characteristic of Geopolymer Based on Lead Smelting Slag"

_ijerph, 2020, doi:10.3390/ijerph17082762_

Round 1
Reviewer 1 Report
The treatment of lead smelting slag is a difficult problem for nonferrous smelting enterprises. The IRLSS based geopolymer was synthesized via water glass activation, which is of great significance of environmental protection and resource recycling. The alkali-activation method is an interesting and useful for other researchers. Therefore, the manuscript is recommended for publication in the International Journal of Environmental Research and Public Heath after the following corrections.
- The English language requires improvement, it is advised to find a native English speaker to edit the paper.
- The water glass (SiO2 ≥ 26.5 %, Na2O ≥ 8.3 %, molar ratio of SiO2/Na2O is 3.3, and density is 1.371 g/cm3)”. Are the components greater than or approximately equal? The content should be clearly verified.
- “water-to-binder ratioon UCS” should be “ratio on”.
- How is the “water-to-binder” ratio defined and operated? The binder is IRLSS, (IRLSS+water glass) or (IRLSS+water glass+water)?
- “with a loading rate of 2.4 KN/S”. Please check the unit format (kN/s).
- The author believes that the excess of alkali causes "frosting phenomenon", which leads to plenty of larger porosity in geopolymer and affect compressive strength. I think this is just one reason. The change in strength may be caused by excess water.
- Only the leaching toxicity of arsenic is increased, is it related to the addition of alkali?
- The research in this article is of great significance for solving solid waste in smelting enterprises. Have your considered using waste raw materials as alkali activators in the future to achieve the purpose of “Using waste to treat waste”?
- Figure 1, the red labels shoud not cover the line, please modify this.
Author Response
Reviewer 1
The treatment of lead smelting slag is a difficult problem for nonferrous smelting enterprises. The IRLSS based geopolymer was synthesized via water glass activation, which is of great significance of environmental protection and resource recycling. The alkali-activation method is an interesting and useful for other researchers. Therefore, the manuscript is recommended for publication in the International Journal of Environmental Research and Public Heath after the following corrections.
Point 1: The English language requires improvement, it is advised to find a native English speaker to edit the paper.
Response: Thank you for your suggestion. We have modified the language.
Point 2: Line 94: The water glass (SiO2 ≥ 26.5 %, Na2O ≥ 8.3 %, molar ratio of SiO2/Na2O is 3.3, and density is 1.371 g/cm3)”. Are the components greater than or approximately equal? Larger is too abstract. The content should be clearly verified.
Response: Thank you for your suggestion. We have replaced by “SiO2 = 26.5 %, Na2O = 8.3 %”.
Point 3: Line 106: “water-to-binder ratioon UCS” may be “ratio on”.
Response: Thank you for your suggestion. There is a space missing. We have added the space.
Point 4: Line 106: “water-to-binder”. How is the “water-to-binder” ratio defined and operated? The binder is IRLSS, (IRLSS+water glass) or (IRLSS+water glass+water)?
Response: Thank you for your question. The binder is (IRLSS+water glass) which is defined in line 109.
Point 5: Line 115: “with a loading rate of 2.4 KN/S”. Please check the unit format (kN/s).
Response: Thank you for your suggestion. We have replaced by the unit format (kN/s).
Point 6: Line 147-148: The author believes that the excess of alkali causes "frosting phenomenon", which leads to plenty of larger porosity in geopolymer and affect compressive strength. I think this is just one reason. The change in strength may be caused by excess water.
Response: Thank you for your suggestion. The coefficient n in the formula Na2O·nSiO2 is called water glass modulus, which is the molecular ratio (or molar ratio) of silicon oxide and alkali metal oxide in water glass. Water content is discussed later.
Point 7: Table 4: Only the leaching toxicity of arsenic is increased, is it related to the addition of alkali?
Response: Thank you for your suggestion. The glass phase is dissociated with water glass, and then Ca2+ and Fe2+ is dissolved to produce C-S-H. During the hydration process, the release of arsenic may be caused by the dissolution of calcium and iron.
Point 8: The research in this article is of great significance for solving solid waste in smelting enterprises. Have your considered using waste raw materials as alkali activators in the future to achieve the purpose of “Using waste to treat waste”?
Response: Thank you for your suggestion. In the future, we will conduct research on alternative waste as an alkali activator.
Point 9: Figure 1, the red labels shoud not cover the line, please modify this.
Response: Thank you for your suggestion. We have modified the labels.
Reviewer 2 Report
The manuscript presents an interesting alternative to use iron-rich lead smelting slag from lead production as raw material for the manufacturing of geopolymers.
I have some questions about the results:
1) Why the UCS tests have been performed only at 3 and 28 days? In my opinion, longer test times, such as 90 or 180 days, would be necessary to test the stability of the matrix, and to probe the compressive strength of the specimens. As authors probably be aware, sometimes when alternative raw materials or additions are used, decreasing of compressive strength takes place at longer times.
2) About effect of modulus of water glass, authors concluded that optimum modulus seems to be 1, however according to the Fig. 2, modulus 1.5 offers almost the same results of compressive strength. Why authors make that decision? Is it because the increasing of frost phenomenon? Is not clear in the text.
3) The resolution of FTIR spectra is not very good, and maybe is not enough to use it to characterize phases. Authors talk about carbonates, but don't specify which kind of carbonates, and why these carbonates have been formed. Please give more information, and discuss more the results.
Author Response
Reviewer 2
The manuscript presents an interesting alternative to use iron-rich lead smelting slag from lead production as raw material for the manufacturing of geopolymers. I have some questions about the results:
1) Why the UCS tests have been performed only at 3 and 28 days? In my opinion, longer test times, such as 90 or 180 days, would be necessary to test the stability of the matrix, and to probe the compressive strength of the specimens. As authors probably be aware, sometimes when alternative raw materials or additions are used, decreasing of compressive strength takes place at longer times.\
Response: Thank you for your question. National standards for cement and other cementitious materials stipulate the need to test the compressive strength of 3d and 28d. The geopolymer material has a fast hardening speed, and the degree of hydration at 28 days exceeds 90%. The strength of the test geopolymers 3d and 28d can represent their early and late strengths. In the future, we will study the compressive strength of hydration for longer time, such as 90 or 180 days.
2) About effect of modulus of water glass, authors concluded that optimum modulus seems to be 1, however according to the Fig. 2, modulus 1.5 offers almost the same results of compressive strength. Why authors make that decision? Is it because the increasing of frost phenomenon? Is not clear in the text.
Response: Thank you for your question. The modulus 1.5 and 1 offer almost the same results of compressive strength. We have modified the conclusion that the optimum modulus of water glass was 1~1.5.
3) The resolution of FTIR spectra is not very good, and maybe is not enough to use it to characterize phases. Authors talk about carbonates, but don't specify which kind of carbonates, and why these carbonates have been formed. Please give more information, and discuss more the results.
Response: Thank you for your question. Calcium carbonate is the main carbonate compound, mainly because the calcium ions in the pore fluid in the hardened body easily react with CO2 in the air to form CaCO3 during the curing process.
Round 2
Reviewer 1 Report
The authors have done reasonable changes to the comments, it is then can be published in the present form.
This manuscript is a resubmission of an earlier submission. The following is a list of the peer review reports and author responses from that submission.